# Aquaporin Gating: A New Twist to Unravel Permeation through Water Channels

**DOI:** 10.3390/ijms232012317

**Published:** 2022-10-14

**Authors:** Marcelo Ozu, Juan José Alvear-Arias, Miguel Fernandez, Agustín Caviglia, Antonio Peña-Pichicoi, Christian Carrillo, Emerson Carmona, Anselmo Otero-Gonzalez, José Antonio Garate, Gabriela Amodeo, Carlos Gonzalez

**Affiliations:** 1Department of Biodiversity and Experimental Biology, Faculty of Exact & Natural Sciences, University of Buenos Aires, Buenos Aires C1053, Argentina; 2CONICET—Institute of Biodiversity and Experimental and Applied Biology CONICET (IBBEA), University of Buenos Aires, Buenos Aires C1053, Argentina; 3Interdisciplinary Center of Neurosciences of Valparaiso, University of Valparaiso, CINV, Valparaíso 2360102, Chile; 4Millennium Nucleus in NanoBioPhysics, Scientific and Technologic Center of Excellence of Science and Life, Santiago 7750000, Chile; 5Cell Physiology and Molecular Biophysics Department and the Center for Membrane Protein Research, Texas Tech University Health Sciences Center, Lubbock, TX 79430, USA; 6Center of Protein Study, Faculty of Biology, University of Havana, La Habana 10400, Cuba; 7Faculty of Engineering and Technology, University of San Sebastian, Santiago 8420524, Chile; 8Department of Physiology and Biophysics, Miller School of Medicine, University of Miami, Miami, FL 33136, USA

**Keywords:** aquaporin, voltage sensor, AQP, water transport, gating mechanism

## Abstract

Aquaporins (AQPs) are small transmembrane tetrameric proteins that facilitate water, solute and gas exchange. Their presence has been extensively reported in the biological membranes of almost all living organisms. Although their discovery is much more recent than ion transport systems, different biophysical approaches have contributed to confirm that permeation through each monomer is consistent with closed and open states, introducing the term gating mechanism into the field. The study of AQPs in their native membrane or overexpressed in heterologous systems have experimentally demonstrated that water membrane permeability can be reversibly modified in response to specific modulators. For some regulation mechanisms, such as pH changes, evidence for gating is also supported by high-resolution structures of the water channel in different configurations as well as molecular dynamics simulation. Both experimental and simulation approaches sustain that the rearrangement of conserved residues contributes to occlude the cavity of the channel restricting water permeation. Interestingly, specific charged and conserved residues are present in the environment of the pore and, thus, the tetrameric structure can be subjected to alter the positions of these charges to sustain gating. Thus, is it possible to explore whether the displacement of these charges (gating current) leads to conformational changes? To our knowledge, this question has not yet been addressed at all. In this review, we intend to analyze the suitability of this proposal for the first time.

## 1. Introduction

Aquaporins (AQPs) are members of a vast and extensive family of transmembrane proteins known as membrane intrinsic proteins (MIPs). Some family members can specifically facilitate the passage of water molecules through cellular membranes (orthodox AQPs), others can combine the passage of water with gases and/or small polar molecules. Since the first water channel was identified [1], a large number of publications have contributed in confirming their ubiquity, diversity and the high impact that many members have on cell physiology [2,3,4,5,6,7]. 

For a long time, before the discovery of the aquaporin family, it was thought that water moves by simple diffusion across the cell membrane. However, water crosses through the lipid bilayers very slowly. Therefore, this idea could not explain the high permeability rates of certain cell types specializing in water transport. On the other hand, the idea that specialized cells had aqueous pores in their membranes that facilitated water movement allowed for an explanation of this physiological phenomenon observed in, e.g., red blood cells (RBC), the urinary bladder of amphibians and the proximal tubules of the nephrons. Thus, based on these experimental observations, Finkelstein (1987) and others argued that it is possible to discriminate between the contribution of specific water channels (pores, now AQP) in a biological membrane considering two parameters: the membrane diffusion permeability (*P_d_*) and the osmotic permeability (*P_f_*). When the ratio *P_f_*/*P_d_* = 1, water is moving by a partition-diffusion process during osmotic events, while when *P_f_*/*P_d_* > 1, water transport is facilitated through pores [1,8]. However, this aqueous pore idea was controversial and caused fervent debate between those researchers who were in favor versus those who believed that water molecules could cross the cell membrane without needing a transport facilitation entity. The historical event that ended this discussion occurred years later (and ruled in favor of Finkelstein) and was the experiments carried out by the group of P. Agre, based on the overexpression of the RBC 28-kD membrane protein (CHIP28, later called AQP1) in *Xenopus laevis* oocytes [1]. The expression of this protein dramatically increased the *P_f_* of the oocytes in osmotic shock experiments, from 0.001 cm s^−1^ to 0.02 cm s^−1^; that is an order of magnitude! Thus, *P_f_* has been considered the key parameter to test the impact of AQPs in a biological membrane. 

Since the discovery of the molecular entity that mediates the water flux in RBD, many research groups undertook to study these proteins, finding and cloning them in multiple species from all living organisms. This leads to addressing questions that naturally arose regarding the structure, the water conduction mechanism and the regulation of water transport. Thus—and mainly—by biophysical approaches, site-directed mutagenesis, structural determinations and molecular dynamics simulations, a huge amount of evidence has been collected to partially answer the questions that ruled the study of aquaporins to elucidate the structure and the mechanisms of water conduction and regulation. 

The large family of aquaporins are all tetramers of four independent water-conducting pores (with homo- and heterooligomers available in certain subfamilies) [9,10,11]. The “signature” channel of each monomer is set by two conserved regions: the NPA-region in the middle of the monomer and the aromatic-arginine (ar/R) motif (also known as ar/R selectivity filter) (Figure 1) facing the extra-vestibular region (Figure 2 and Figure 3). Each monomer also possesses certain symmetry resembling an “hour-glass” pore structure, forming a single-file narrow water-conducting pathway [12,13,14,15] (Figure 3). These structural features are highly preserved in AQPs and distinguish them from many other membrane channels. For example, ion channels, in general, do not share this symmetry about the mid-plane of the membrane and also do not generate four pores but only one conducting pathway (Figure 2) [16]. Of course, there are exceptions. In mammals and other animals, the Hv1 proton channel can be found as a dimer, but each monomer can function independently as its own voltage sensor and conduction pathway [17,18,19]. It is postulated that Hv1 from some unicellular organisms lacks the predicted coiled-coil regions sustaining dimerization and function as monomers [20]. Thus, while in proton channels, oligomerization is optional, with functional implication, it seems to be mandatory in AQPs. Although in AQPs each monomer is a functional channel [21] and separated monomers can be reconstituted in liposomes in vitro [22,23], in nature, four water channel subunits should come together to form a tetramer in the membrane [24]. 

Despite very specific exceptions that describe ion conductance [28], orthodox aquaporins are electrically silent [29,30] unless activated (See Section 4). Thus, while the study of ion channels’ conduction properties can be achieved by electrophysiological methods, the study of the water transport capacity of AQPs relies on complementary biophysical approaches. This is a difference of study capacities between ion and water channels. Electrophysiological methods allow the direct recording of ion fluxes through the channels (macroscopic currents or single channel recordings) and even the charge displacement in particular regions of the channel during the transition from closed to open states (gating currents). Thus, in the field of ion channels, electrophysiological methods can provide direct information at the single channel level. For example, studies performed by patch-clamp and voltage-clamp fluorometry demonstrated that the most consistent functional difference between monomeric and dimeric proton channels is found in gating kinetics [31,32,33,34]. On the other hand, there is no direct technique to achieve this level of information in the field of AQPs because water fluxes cannot be measured directly. Only the consequences produced by water fluxes can be measured, i.e., volume or concentration changes.

In aquaporins, the term gating as used to describe a mechanism triggered by phosphorylation, pH or Ca^2+^ comes from a series of works emerged from the crystal structure of the plant aquaporin SoPIP2;1 in open and closed conformations [35], together with physiological evidence [36,37] and molecular dynamics (MD) simulations [38]. Other regulatory mechanisms were described in AQPs, for example, heterotetramerization or expression regulation at nucleotide and protein levels [39,40,41,42]. Certain plant AQP-PIP subfamilies increase water permeability when present as heterotetramers relative to the homotetramers [43,44,45,46]. In addition, a cooperative response of the four monomers has been proposed [47] and experimentally explored in different aquaporins [48,49,50,51]. However, as occurs in ion channels, the gating of AQPs provides the fastest response to abruptly modify the water membrane permeability. 

The study of ion channels combines experimental and simulation methodologies. The former are powerful, while the latter are very limited due to the lack of high-resolution structures. The opposite occurs in the field of AQPs; there are no direct experimental methods at the single channel level but there are many structures resolved with very high resolution (see Table 1). This, plus the relatively small size of AQPs, sustains molecular dynamics simulations with high predictive power. Therefore, the unitary study of the protein—performed by patch-clamp in ion channels—is constricted so far to molecular dynamics approaches in AQP. Given the fast kinetics (in the range of nanoseconds) and the discrete nature of particle fluxes in confined geometries, MD has arisen as an ideal tool to scrutinize AQP’s function at the atomistic level. Alternative to the lack of experimental demonstration, molecular dynamics simulations performed by de Groot and coworkers [52] confirm the single-file fashion transport hypothesis postulated before the discovery of AQPs [53]. 

Indeed, other molecular dynamics simulations carried out by de Groot [85] demonstrated the role of the NPA region as an electrostatic cap that blocks proton transport but allows water passage. Since then, a great deal of MD studies had characterized water transport ([86] and references therein), and even simplified models based on carbon nanotubes have been shown to qualitatively reproduce AQPs [86,87,88,89].

In the following sections, we will explore the term gating used in the AQPs field and discuss the implications of introducing new experimental approaches to unravel its mechanism. The discussion will be focused on aquaporins that mainly transport water, at both the experimental and theoretical levels.

## 2. Aquaporins: Structure and Function

Aquaporins are small and very hydrophobic members of the MIP superfamily, with predicted sizes that range from 27 to 31 kDa (200 to 300 residues). Those that are exclusively (or mainly) water-selective mediate the bidirectional water flow driven by an osmotic gradient but block proton transport [1,22,90,91]. In detail, the proton blockage is achieved due to the presence of an electrostatic barrier that highly penalizes the proton desolvation [63,73]. Water crosses in a single-file fashion, performing a dipolar rotation of water molecules in the highly conserved NPA region (Figure 3), thus impeding a Grotthuss-type proton hopping [52,73,85,92].

Unlike ion channels, the pore for the passage of water does not reside in the center of the tetramer, but each monomer constitutes an independent water pathway [93]. The sequence alignment of aquaporins (Figure 1) shows several highly conserved motifs [93], including the signature sequence motifs of these proteins, i.e., two repeating Asn-Pro-Ala or NPAs [12]. The conformation of each monomer in the membrane shows six membrane-spanning α-helices in two tandem repeats (each one formed by three transmembrane domains) and two conserved long loops, one cytosolic (loop B) and the other extracellular (loop E), both of which contain one NPA motif and a short α-helix (Figure 3). These loops deepen into the membrane opposing the two NPA motifs in the center of the channel and exit the membrane from the same side by forming a short α-helix that contains a highly conserved arginine residue. This arginine faces a histidine located in the fifth transmembrane segment. Both of these residues conform the highly conserved aromatic/arginine (ar/R) region, which faces the extracellular side and constitutes the selectivity filter (Figure 3) [85,94]. Both the N and C termini of the proteins are intracellular.

The structure of aquaporins shows particular symmetric features. In general, each AQP monomer is 60 Å in height and has a diameter of 30 Å, with the tetramer covering a surface of about 60 × 60 Å^2^ [95]. Furthermore, each monomer also possesses certain symmetry, resembling an “hour-glass” pore structure. Both the cytoplasmic and the extracellular entrances have a conical-type shape and continue to the single file region located at the center of the membrane [12,13,14,15]. This structural design assures highly permeable water channels capable of translocating 10^9^ H_2_O molecules s^−1^ (per monomer) in a single-file regime [92,96,97], similar to other apolar channels such as carbon nanotubes [98,99]. This fast transport behavior is the result of the unusual properties of 1D water enclosed within hydrophobic walls. In these conditions, the almost frictionless flux plus the highly correlated motion due to the aligned dipoles lead to the observed and predicted high permeability [85].

Interestingly, MD simulations of simplified AQP models have shown that the four moving water files do generate a friction effect that reduces the flux across the channels [88]. This effect is in accordance with the observation that the number of H-bonds the water molecules form (and immediately break) with pore-lining residues during the passage through the single file region correlates with the water transport capacity of the channel, i.e., more H-bonds produce lower permeability [100].

To date, the function of aquaporins is characterized by the osmotic permeability coefficient (*P_f_*) of the membrane where the channel is located. However, experimental methods to determine *P_f_* make it nothing else than a parameter that characterizes the membrane as a whole. Unfortunately, unlike ion channels and the patch-clamp technique, unitary water permeation through a single AQP cannot yet be unequivocally measured by means of a specific technique. Determinations of the unitary permeability coefficient (*p_f_*) from experimental or MD simulation approaches are in the range of 1 to 10 × 10^−14^ cm^3^ s^−1^ [96,97,100,101,102,103,104,105,106,107,108,109,110,111]. Although accuracy on the determination of the single channel permeability coefficient has been improved [112], this is still an indirect calculation derived from *P_f_* and channel density determinations. This means that there is a long way to go from studies of *P_f_* to finally obtain direct experimental measurements of the events occurring to water molecules inside the channel. This is a question that has not yet been approached due to the lack of experimental methodologies.

To date, the crystal structure of AQPs from more than ten different sub-families has been resolved (see Table 1). As can be seen in Table 1, the crystallographic resolution is very high, so that in some cases the electronic densities of the oxygen of the water molecules could be resolved inside the water channel [63]. The invaluable number of high-resolution X-ray structures of aquaporins from different living organisms allows: (i) the identification of those basic structures shared in common, (ii) unraveling to a molecular level function-structure features, (iii) the elucidation of the sequence of events that are requested to achieve mechanisms as gating, provided the availability of open and closed states of the same aquaporin. These three features are interconnected (or interweaved) by a subject that has not been revealed yet in AQP: the gating of currents enough to sustain a gating mechanism. In particular, three sites (loop D, the ar/R region and loop B) show highly conserved residues, and some of them could be considered critical for sustaining our working hypothesis.

## 3. Gating in Plant and Animal Aquaporins

Membrane water permeability can be modified in different ways. Regulation mechanisms were described in detail at different organization levels. At the cellular level, expression and trafficking of new synthesized AQPs or from pool reservoir vesicles are well known both in animals and plants [113]. At the level of the tetramer, hetero-oligomerization modifies the water transport capacity of the conforming subunits [11]. At the molecular level, the best-known gating mechanisms are mediated by pH, or calcium-binding and phosphorylation, which are also related to traffic events [7]. Other gating mechanisms were discovered but are still not elucidated (e.g., mechanosensitivity) or are controversial, such as the capacity of AQPs to transport ions [114].

If the traffic mechanism depends on the synthesis of new AQPs, the regulation of membrane permeability is slow (in the order of hours). However, if traffic events involve only the channels that are pooled in cytosolic vesicles, then changes in membrane water permeability are faster (in the order of seconds). However, in gating mechanisms at molecular level, i.e., conformational changes that modify the inner water-transport capacity of the channel are even faster (in the order of milliseconds). In this review, we make a brief revision of the regulation mechanisms described in AQPs and focus on gating mechanisms that directly affect the water pathway of AQPs at the molecular level.

### 3.1. Trafficking of Aquaporins

Subcellular localization of AQPs is diverse. Some members are intracellular and others are localized in the plasma membrane. The localization of intracellular mammal aquaporins such as AQP11 and AQP12 depends on the sequence of the NPA motif [115,116]. In plants, PIP2-type aquaporins are usually expressed in the plasma membrane while PIP1-type are usually retained in the endoplasmic reticulum (ER) [117]. However, PIP1 members can reach the plasma membrane by forming heterotetramers with PIP2 subunits. Oligomerization occurs in the membrane of the endoplasmic reticulum, and a diacidic motif (DXE) necessary for proper trafficking [117] would be part of an ER-sorting checkpoint after the formation of homo- or heterotetramers [118]. Then, the different combinatory possibilities for heterotetramerization produce tetramers with different water transport capacity. This mechanism is highly preserved in PIP members and was probed in vitro by heterologous expression in Xenopus oocytes [43,45,46,119,120] and in vivo in maize cells [117]. This common mechanism in PIPs is rare in animal aquaporins. Only AQP4 relocalization seems to be related to heterotetramerization. The AQP4 M1 and M23 isoforms have different capacities of diffusion and binding to adhesion complexes [10]. In consequence, M23 can not only form orthogonal arrays and have different localization than M1 but also, as well as PIPs, can form heterotetramers with M1 subunits [10]. 

It is well documented that relocalization of AQPs is dependent on phosphorylation events. Subcellular localization of mammalian AQPs is mediated by several protein kinases pathways and would be cell-type specific [113,121]. In AQP0, phosphorylation of Ser235 immediately after biosynthesis is required for correct trafficking to the plasma membrane [122]. In AQP2, phosphorylation of different serine residues redirects the channel to a different membrane. Phosphorylation of Ser256 is necessary for targeting AQP2 to the apical membrane [123]. Once in the apical membrane, phosphorylation of residue 269 increases the retention time at the membrane [124]. Other serine residues can also be phosphorylated in AQP2. When Ser261 is phosphorylated, AQP2 is retained in storage vesicles [125,126], and phosphorylation of Ser264 is related to exosome excretion of AQP2 [127]. The localization of other mammalian AQPs is also dependent on phosphorylation events and is related to diseases (Table 2 summarizes this information). We only mention a few examples in our paper; detailed revisions on phosphorylation and trafficking in mammalian aquaporins were recently published [113,128].

### 3.2. The Well-Defined Role of Cytoplasmic Loop D in Plant PIPs

A highly conserved group of plant aquaporins known as PIPs (mainly located in the plasma membrane) has been extensively studied for its capacity to rapidly adjust membrane water permeability. Dephosphorylation of specific serine residues [35,146,147,148], cytosolic acidification [36,45,82,120,149] and intracellular calcium depletion [37,150,151] are all conditions that favor the closed state of the channel. Thus, the three cases are considered as triggering a gating mechanism in a completely reversible manner. Here, we will focus on the one mediated by pH, where structural biology and MD simulation information fits with the complemented experimental findings. 

A convincing biophysical experiment of aquaporin gating came from studies performed in an enriched fraction of root plasma membrane vesicles from *Beta vulgaris*, confirming that the high water-conductance of a purified fraction of plasma membrane vesicles could be effectively reduced by 98% when cytosolic pH was dropped [37]. As expected, the inhibition also produced an increase in the activation energy for water transport, confirming channel closure.

The pH-regulation of different plant species was confirmed for PIP aquaporins heterologously expressed in *Xenopus* oocytes [36,45,120,149,152]. In particular, mutations of a strictly conserved histidine residue in loop D of AtPIP aquaporins—which corresponds to His193 in the spinach plasma membrane aquaporin SoPIP2;1—confirmed that pH sensitivity depends on the protonation state of this conserved residue [36]. Additionally, a different aquaporin (a TIP, VvTnTIP2;1) with His131 mutated to aspartic acid (D) or alanine (A) also resulted in the loss of a pH-dependent decrease in water permeability when overexpressed in yeast [153].

The molecular mechanism of gating by pH has been elegantly described in SoPIP2;1 from *Spinacia oleracea* from crystallization performed in its open state at physiological pH and at the closed state with cytosolic acidification [35,82]. Molecular dynamics simulation suggests that in the transition from the open to closed state, the intracellular loop D moves towards the pore entrance concomitantly with a vertical displacement of the fifth transmembrane segment. These movements imply a displacement of about 8 to 15 Å for Leu197, Pro195 and Val194, which are in loop D.

In the closed state, loop D blocks the intracellular pore entrance, and this configuration is stabilized through a network of hydrogen and ionic bonds. Arg190 and Asp191 (both from loop D) form hydrogen bonds with Gly94 (from loop B) and Ser36 (TM1). In addition, hydrogen bonds between a water molecule and His193 (loop D), Ser115 and Lys113 (latter both from loop B) contribute to the stabilization of the closed state. For the interaction between His193 and Ser115 to occur, the double protonated state of His193 is needed. Since the pK_a_ for Ser115 is near pH 6, the interaction that stabilizes the closed state occurs at this pH or lower.

Since Ser115 could be phosphorylated or dephosphorylated, then the water channel SoPIP2;1 can also be gated by phosphorylation–dephosphorylation events. It was demonstrated that the channel closes with dephosphorylation of Ser115 and Ser274, located in loop B- and C-terminal, respectively [146] (Figure 1).

### 3.3. The Role of Histidines in Animal Aquaporins

The pH regulation was described in certain animal AQPs. However, the mechanism seems not to be conserved as in plant PIPs. In AQP6 and AQP0, lowering pH induces a totally opposite effect: an increase in water transport capacity [154,155]. The pH sensitivity of AQP0 is affected by mutations on His40 from the extracellular loop A [155]. The involvement of histidines from loops A and C on pH dependence was also confirmed in other aquaporins such as BtAQP0, MIPfun, HsAQP1 and RsAQP4 [156]. 

This evidence refers to the role of extracellular loops A and C, as well as evidence in plants referring the role of intracellular loop D.

On the other hand, a recent study clearly showed, theoretically and experimentally, that h-AQP4 possesses a pH sensor within the plane of the membrane [26]. By MD simulations, the protonation of His95 (located at the cytoplasmic entrance of the single file region) correlated well with a local pore radius increase and predicted enhanced permeability. The latter was confirmed by a gain of water influx of AQP4-expressing *Xenopus* oocytes challenged with a hyperosmotic gradient after exposure to intracellular acidification. This result is an illustrative example of how very subtle changes (a single proton leads to an increase of less than an angstrom in the pore radius) can lead to macroscopically measurable effects, showing how MD is an excellent tool that can trace and predict the molecular underpinnings responsible for the macroscopic observed phenomenon.

### 3.4. The Cytoplasmic Loop B: More Critical Motifs

Loop B, which contains the first NPA motif, shows a highly conserved sequence just before the NPA constituted by two glycine residues flanking three other residues that, in most cases, are H, I and S (Figure 1). The GHISG sequence located in this loop is notable since GxxxG sequences constitute the signature motif of transmembrane segments [157]. Although both loop B and E deepen into the membrane, creating a seventh pseudo transmembrane segment, the GxxxG sequence before the first NPA does not constitute an α-helix. However, G and H residues create the mouth of the single file region (Figure 3) and are crucial for water passage through the channel [57,63]. Even more intriguing, it was proposed that GxxxG sequences are also the signature motif of mechanosensitive ion channels [158]. In aquaporins, water translocation through the channel of bacterial, animal and plant members can be regulated by membrane tension changes [159,160,161,162]. Experimental and simulation results performed with AQP4-reconstituted liposomes with phospholipids of different lengths [163] invite us to suppose that a conformational change could occur in the cytoplasmic mouth of the channel when the membrane is stretched [114]. This would constitute a gating mechanism involving G and H near the first NPA motif. It is worth noting that in this structural region, H95 is also located, just between the GxxxG and the NPA sequences in h-AQP4. Since the G and H of this region are involved in the formation of H-bonds with the water molecules that enter the single file region [100], a conformational change in the cytoplasmic mouth could affect the collective motion of the water file into the channel. Although these reports invite us to imagine a gating mechanism, direct measurements to confirm these hypotheses are lacking.

### 3.5. The ar/R Selectivity Filter and More Charges Available

The selectivity filter is localized about 4 to 8 Å towards the extracellular side from the NPA region. This filter is located at the extracellular entrance of the single file region (also within the membrane plane) and determines selectivity by size [94]. In aquaporins that exclusively transport water, the ar/R region contains a Phe residue from TM2, a His residue from TM5 and two residues from Loop E: one small residue that provides a backbone carbonyl oxygen, usually Cys, Thr or Ala, and an Arg that is highly conserved in both water-transporting aquaporins as well as aquaglyceroporins (Figure 1). In the aquaglyceroporins, the Phe from TM2 is replaced by a Trp residue, the His from TM5 is replaced by a Gly residue and the small residue from Loop E is replaced by a Phe, giving rise to a wider selectivity filter [102,164,165,166,167].

In orthodox aquaporins, both the His from TM5 and the Arg from loop E provide donor hydrogen bonds for water molecules, and their position as well as the distance between them is crucial for ordering water in the extracellular aperture, just at the end of the single file region [57,85]. Interestingly, molecular dynamics simulations predict that both R195 of bovine AQP1 and the homolog R216 of human AQP4 can adopt different positions driven by transmembrane voltage differences [168]. The various positions adopted by these residues in the simulations coincide with those observed in crystals [58,63]. In AQP1, the distances between R195 and H180 showed a wide distribution for this pair of residues [168], with the two main positions of the R being coincident with observations in crystals [57]. In AQP4, the distribution of distances between R216 and H201 is related to the position of the guanidium group of the R216. The up position is associated to a possible open state, while the down position occludes the pore and is associated to a possible closed state [168]. In these simulations, the movement of the positive charge of the guanidium group is driven by positive and negative electrical potentials, which suggests that AQP1 and AQP4 can be gated by the membrane potential. Although these are stimulating observations, the simulations were performed with very high electrical potentials: 1500 and –1500 mV, which exceeds physiological values by far [168].

The usage of physiologically high membrane potentials is usual when studying water transport across narrow pores, such as nanotubes and AQPs. The previous is usually performed in MD to reduce the kinetic barriers of rare events, e.g., the opening of a channel, which at current MD capacities (now routinely carried out in the range of hundreds of nanoseconds or several microseconds) are hardly sampled at physiological potentials. Nevertheless, the above disturbs water dynamics and thermodynamics [169]. Moreover, within confined geometries, the bulk shielding effect is no longer present, further increasing the field effects [88]. Therefore, the usage of lower voltages is not only necessary to approach physiological conditions but also to disturb the dynamics and thermodynamics of the water dipole as little as possible.

An illustrative example of how water is disturbed by external voltages is shown in Figure 4. In detail, the free-energy profile is projected along the cosine of the angle formed by the dipole vector and the *z* axis, i.e., water orientation is shown for different field strengths. For zero field conditions, a flat profile is present; for a physiological membrane potential (around 80 mV), differences are below thermal noise (k_B_T = 0.6 Kcal mol^−1^ at 298 K) (Figure 4A). For higher potentials (around 200 mV), differences are above k_B_T; in other words, from here on, it starts to be thermodynamically relevant. The latter affects H-bonding capacities, diffusivity and orientational relaxation times, among others [169]. Thus, lower voltages than the ones previously employed [168] should be utilized when studying water transport to avoid these problems. A consequence of the latter is that the signal-to-noise ratio in MD simulations is decreased, thus requiring longer simulations to reach convergence and statistical certainty [170].

In this way, MD studies of human AQP4 performed with electrical potentials closer to physiological membrane values predict that hyperpolarizing voltages trigger fluctuations in the side-chain orientation of H201 at the selectivity filter. Nevertheless, these events would not render any increment/decrease in water permeability [103]. Moreover, the membrane potential tested was still way behind physiological conditions (–520 mV) but lower than what was previously reported in the literature [168]. Intuitively, it is often thought that charged residues are the responsive elements towards external fields [172]. However, at these lower voltages, R216 did not experience any noticeable change [103]. On the other hand, free-energy estimations of H201 side-chain orientation revealed a change in the torsional thermodynamics upon a depolarizing potential of 260 mV [173]. This is of great importance as, even though still away from cellular conditions, this voltage difference is applicable in patch-clamp experiments.

Microsecond MD simulations of polarizing and depolarizing potentials predicted the existence of two gates: the upper (H201-R216) and lower ones (H95-C178) leading to four states in each monomer [170]. In this work, the depolarizing potential shifted H201-R216 distance distributions towards higher values (around 6.5 Å), termed “aggressively open states”, and were associated with a voltage-dependent gating phenomenon in h-AQP4. In doing so, free-energy estimations revealed the existence of a bimodal mode for the H201 side-chain orientation profile that was altered for depolarizing potentials. Intriguingly, even for these relatively long MD simulations, again, R216 did not show any noticeable response towards these potentials.

Overall, these series of MD studies are consistently suggesting that: (i) H201 is a flexible hinge presenting a bimodal state with low kinetic barriers and (ii) depolarizing potentials lead to a change in the orientational thermodynamics of H201 that slightly increase the selectivity filter size. Interestingly, the fact that depolarizing potentials seem to render a more open channel implies that the default setting for human AQP4 is to be more closed [170].

Evidence revisited up to here gives rise to the question: Are certain AQPs voltage-dependent channels? This would imply that a voltage sensor is present in aquaporins and that the channel opens with depolarizing voltages. However, conclusive evidence still needs to be collected to (i) demonstrate the voltage-gating phenomenon in human AQP4 and, (ii) more importantly, whether it correlates into function or, in other words, changes in water permeability [26]. The former can be achieved in MD with cleverly designed free-energy calculations in which the gating process is projected along a judiciously chosen reaction coordinate, e.g., the four dipoles of H201, R216, H95 and C178. However, the experimental measurement of gating currents by patch-clamp could be a technical challenge as this potentially involves the displacement of a single residue (H201) which is not necessarily charged. The latter, from an MD approach, will require simulations with the gates restrained in the open/closed, properly weighted by their corresponding free-energy profiles; experimentally, patch-clamp studies of cells in hyper osmotic conditions must be carried out. 

## 4. Electrophysiological Studies in Ion Conducting Aquaporins (icAQPs)

The first ionic conductance in AQPs was reported in hAQP1-expressing *Xenopus laevis* oocytes [174]. The authors reported a non-selective cationic conductance that was activated by forskolin and diminished by the application of HgCl_2_. Although this work was questioned at first [175], posterior works reported that AQP1 reconstituted in lipid bilayers showed large single channel conductance activated by cGMP [176], as well as subconducting states [177].

A displacement of loop D seems to be required for the activation of ion currents by cGMP in AQP1, since mutations in this loop impair the development of macroscopic currents [178]. How the displacement of loop D can be associated to gating events occurring in the single file region of the channel is unknown. Indeed, since both the NPA and the SF regions act as a concerted filter blocking the passage of cations in orthodox AQPs [179], it is thought that ion conduction in AQP1 occurs through the central pore formed by the four monomers and not through each of their water pathways. The hypothesis that cations permeate through the central pore of AQP1 is supported by molecular dynamic simulations [178], site-directed mutagenesis experiments [180] or by employing specific blockers [181,182].

Ion current recordings were also reported in AQP6-injected *Xenopus* oocytes. These currents were registered by two electrode voltage-clamps after stimulation with HgCl_2_ or acidification of the extracellular medium [154]. The results published in this work show that the ion conductance of AQP6 is independent from the membrane voltage, at least in the tested range. In addition, single channel currents were registered by using the patch-clamp technique in oocytes with AQP6 [25]. Based on the results obtained with mutants, the authors proposed that each monomer has an individual permeation pathway for ions that is modulated by C155 and C190. The authors suggest that Hg^2+^ traps (i.e., maintains) AQP6 in a conformation that enhances both cationic conduction and water permeability [25]. Since C190 is near the SF (see Figure 3), then, the conformational change associated with the trapping of AQP6 could be affecting the relative positions of both key residues of this region: the highly conserved arginine and histidine.

Unlike with AQP1, cationic currents through AQP6 suggest that positive charges are not repelled at the NPA. Therefore, the ion conduction pathway would be the monomer and not the central pore.

The evidence suggests that animal AQPs can conduct ions if activated. Activation stimuli are diverse: HgCl_2_, cGMP or extracellular acidification. However, it is still unknown which is the ion pathway. Experimental evidence is lacking to know if the ion pathway is the same as the water pathway, or if it lies in another site of the monomer, or if it is structured with the four monomers, e.g., the central pore of the tetramer.

On the other hand, in some plant aquaporins, water transport is not selective, allowing the passage of multiple solutes such as CO_2_, H_2_O_2_ and even ions such as Na^+^ and K^+^. For example, AtPIP2;1 was reported to conduct non-selective cationic currents triggered by intracellular acidification as well as increase intracellular Ca^2+^ concentration [27]. AtPIP2;1 shows no-selective cationic currents with the same regulatory mechanisms as for water permeation, suggesting that cations and water share the same pathway [27,183], supporting the monomeric pathway hypothesis. Evidence based on a natural mutation on G100 of VvPIP2;5 supports this hypothesis. Molecular dynamic simulations performed with a mutant (G100W) of VvPIP2;5, [152], as well as site-directed mutagenesis (G103W) of AtPIP2;1, confirmed that this mutation impairs water permeation and ion conductance but does not affect the translocation to the membrane [27,184]. It is interesting that this glycine residue is in loop B, before the first NPA, and just in the cytoplasmic extreme of the single file region (Figure 3). In addition, Na^+^ and K^+^ conductance was observed in AtPIP2;1 when found in phosphorylated states [185]. In this report, it was observed that after phosphorylation and during ion conduction, the osmotic permeability is reduced while ionic permeability increases. This observation suggests that AtPIP2;1 equalizes its permeability behavior by phosphorylation to support different processes depending on its physiological conditions.

If macroscopic (or even single channel) currents can be registered in experiments with AQPs (understanding that theses currents are due to the passage of ions through the AQP and no other channel), and if these currents are activation-dependent (by cGMP in AQP1, Hg^2+^ or pH in AQP6, or Ca^2+^ or phosphorylation in AtPIP2;1), then a conformational change must occur. Furthermore, if these conformational changes are associated or imply the movement of charged residues or dipoles, it could be measured as gating currents, as in voltage-gated ion channels [186] or Hv1 [19,187]. The location of key residues (C190 in the SF of AQP6 or G103 in loopB of AtPIP2;1) is suggestive. These regions constitute the extracellular and intracellular extremes of the single file region (Figure 3), i.e., both are in the core of the membrane, are key for water permeation and both have residues with positive charge: the conserved Arg and His (depending on its protonation state) in the SF and a conserved His (depending on its protonation state) at the cytoplasmic extreme of the single file region. Finally, it was reported that in AtPIP2;1-AtPIP1;2 heterotetramers, the ion transport is impaired but not the water permeability [27]. This, plus the extended evidence on regulation by heterotetramerization [11,45], suggest that conformational changes in homotetramers and heterotetramers must be different. Thus, if gating currents can be measured, then it can be hypothesized that they could be different between homo- and heterotetramers.

## 5. How Is AQP Sensing the Membrane Electric Field?

Different AQP motifs have a role in regulating the conduction of H_2_O and its properties. It was mentioned that loop D in PIPs and loop B in animal AQPs are related to positive or negative modulation of the pH-mediated gating. On the other hand, in the H_2_O permeation pathway, the ar/R motif is the main filter both by size and by solute load. In the center of the membrane, the NPA motif is critical for the rotation of the water dipole, which enables bidirectional conduction. In addition, it constitutes the first energy barrier to prevent the passage of protons, since the rotation of the angle of the water molecule hinders the formation of hydrogen bonds for the transport of protons.

The results mentioned in the previous sections suggest the possible existence of a region in the protein that is sensing the potential of the membrane. Possible conformational changes of the protein related to either the movement of charged residues or dipoles that are part of the structure of the channel could somehow be correlated with the movement of water through the protein, which could have functional repercussions on the conduction mechanism. To better visualize this point, an interesting strategy is to evaluate the structure –function relationship of AQPs with that of proteins specialized in the sensation of membrane potential, such as voltage-gated ion channels. 

The superfamily of voltage-gated ion channels (VGICs) consists of proteins that undergo conformational changes in response to membrane potential differences. These molecular sensors are like an ON/OFF switch that allows the opening and closing of an ion conduction pathway through the membrane, which, depending on the protein channel, corresponds to a tremendously efficient transport [188]. This capability is conferred by charged residues or dipoles in the protein structure located in the center of the membrane that move in response to changes in the membrane electric field [186,189]. 

The *Drosophila melanogaster Shaker* K^+^ channel was the first K^+^ channel cloned [190]. It has been widely studied from structural, functional, pharmacological and physiological points of view. The vast majority of pioneering research in the field of VGICs was performed in this protein, thus, it was reinforced over time as the paradigm of voltage sensors. *Shaker* K^+^ channel is a tetrameric protein whose subunits consist of six transmembrane segments (S1-S6) that possess two critical domains: the pore domain (PD) and the voltage sensor domain (VSD). The PD is a structural region (composed of S5–S6 segments) that constitutes a functional conduction pathway when each subunit is assembled in the oligomeric structure of the protein (Figure 2). On the other hand, the VSD is a region that confers voltage sensitivity, and it is constituted by the first four transmembrane segments [186]. The voltage sensitivity of the vs. is produced by the presence of four arginine residues in S4 [189]. These charged residues are located in the middle of the membrane and move according to the electric field, inducing a voltage-dependent conformational change. This conformational change of the VSD is mechanically coupled with the PD, facilitated by a motif called gating charge transfer center (GCTC) that leads to the opening of the channel (electro-mechanical coupling) [191]. It is worth mentioning that segments S1–S2–S3 contain highly conserved negatively charged amino acids (aspartate and glutamate) as well as highly conserved aromatic amino acids (tryptophan, phenylalanine or tyrosine). These conserved residues appear to be involved in the stabilization of S4 in different conformational states.

When the literature alludes to some *Shaker*-like channels, it refers to a voltage-gated ion channel having the structural features mentioned above. Despite this, there are ion channels that come out of the paradigm, such as the voltage-gated proton channel, Hv1. The Hv1 channel is a proton selective channel composed of four transmembrane segments (S1–S4). Making a structural comparison with the structure of the *Shaker* K^+^ channel, Hv1 lacks the canonic pore domain, thus, proton permeation necessarily occurs through the voltage sensing domain [192,193]. Its S4 contains 3 Arg residues that confer voltage sensitivity [32,194]. Its oligomeric state is a homodimer assembled by a coiled-coil interaction between C-terminals of each subunit [17]. However, dimeric assembly is not required to form a functional protein. Prevention of subunits interaction (ΔNΔC construct) results in a completely functional voltage-gated proton selective channel with faster activation kinetics than the dimer [17,18,32,195]. Having said the above, the only common feature that *Shaker* and the Hv1 channel probably share is the voltage sensor arrangement of the S4 segment that consists of arginine residues at every third position [189]. 

Movement of charged or dipole particles produces a current. Thus, the signature of every voltage-dependent activation is a transient current produced by the movement of the charged side chains of the confined residues in S4 across the membrane in response to a change in the membrane potential [186,196,197,198]. These currents are known as gating currents and are directly detectable by using the voltage-clamp technique (Figure 5). Studies of gating currents by electrophysiological approaches have helped to reveal the mechanisms of opening, closing, inactivation, drug sensitivity and pH sensitivity, among other key properties of these proteins, being a key experimental approach to the understanding of the structure and function of voltage sensors. 

Even recently, in the monomeric voltage-dependent proton channel Hv1, it was described that the sensitivity of changes between intracellular and extracellular pH (ΔpH = pH_int_ − pH_ext_) is due to the transmembrane segment S4, which is also its voltage sensor [187]. This mechanism would be related to the increase in water permeation events that would facilitate the movement of protons in the active state of the channel, allowing the proton gradient to be dissipated through the movement of the S4 sensor. This could indicate that the voltage-gated proton channel Hv1 is a bridge between the conduction mechanism of “classical” ion channels and the aquaporins. Water channels could be a different evolutionary pathway of the VSD motif, provided the conserved residues we have described here are potential candidates for sensing the membrane voltage and allowing a change in the state of the channel. 

## 6. Perspectives

Although experimental evidence allows the assertion that aquaporins are gated by several mechanisms, and molecular dynamics simulations predict the conformational changes related to those mechanisms, these changes have not been measured experimentally within the membrane plane, thus far.

As was revisited here, the well-described gating mechanism mediated by pH in SoPIP2;1, which is conserved in other PIPs, involves the displacement of intracellular loop D. On the other side of the membrane, extracellular loops A and C are related to gating mechanisms in animal AQPs. However, these events occur outside the membrane plane. Concerning the single file region of the water pathway, several events related to both *p_f_* and *P_f_* changes were described by molecular dynamics simulations as well as suggested by indirect experimental approaches. Unfortunately, direct experimental evidence to confirm these events occurring in the core of the membrane plane is lacking.

We propose that the displacement of key amino acid residues located in the single file permeation pathway is related to gating mechanisms and can be measured by means of the same methodology used on proton channel (Figure 6). The displacement of those residues constitutes transition events among different structural states that would be related to different water transport capacity. To date, distinct states of the aquaporin structure have been described by x-ray diffraction and Nuclear Magnetic Resonance methods. In addition, molecular dynamics simulation predicts the movements of some residues related to such states. However, the dynamic transition between those states has not been measured yet. We propose that the conformational changes related to those states can be measured by electrophysiological methods. In addition, the transmembrane potential needed to register these displacements as gating currents would not be so high as MD simulations performed by Hub and coworkers [168] but most similar to those predicted by Reale and coworkers [173], perhaps around 300 mV. Such measurements will open a new framework of study, improving the capability to elucidate the structural changes of the channel during the transition between open and closed states.

## Figures and Tables

**Figure 1 ijms-23-12317-f001:**
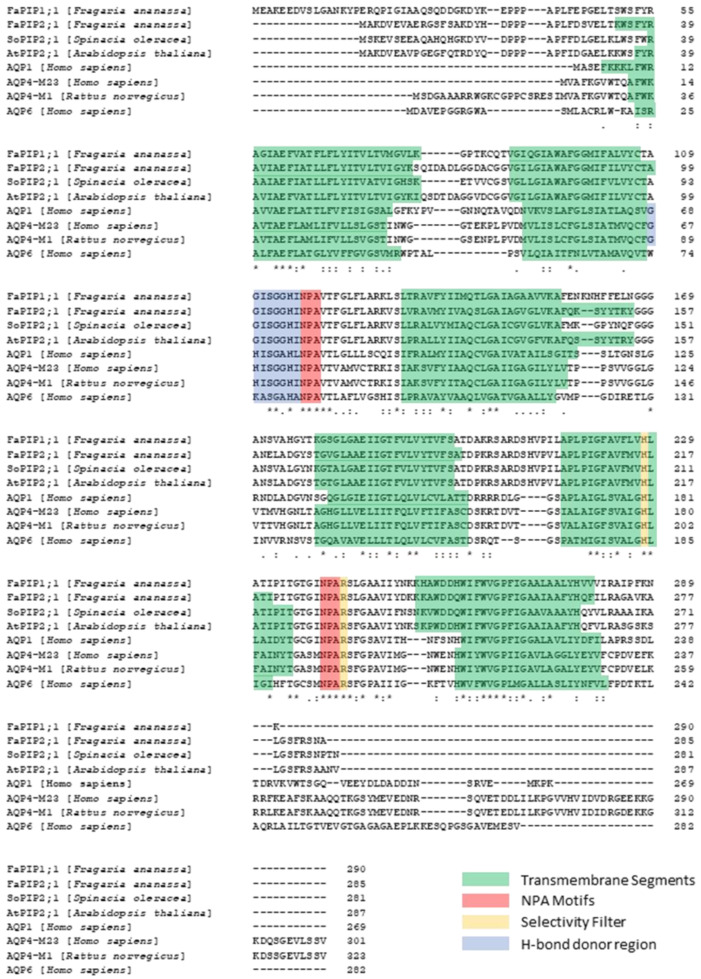
**Sequence alignment of aquaporins from different species.** Multiple sequence alignment by Clustal Omega The alignment of different water-selective aquaporins from plants (*Fragaria ananassa*, *Spinacia oleracea* and *Arabidopsis thaliana*) and animals (*Rattus norvegicus*, and *Homo sapiens*) shows highly conserved residues that are critical for the selective conduction mechanism of water molecules. The transmembrane segments are highlighted in green, the NPA motifs in red, the Ar/R selectivity filter in yellow (His and Arg residues) and the H-bond donor residues of the cytosolic aperture in blue. Alignment symbols: (.) Aminoacid with weakly similar properties. (:) Aminoacid with strong similar properties. (*) Single conserved aminoacid residue.

**Figure 2 ijms-23-12317-f002:**
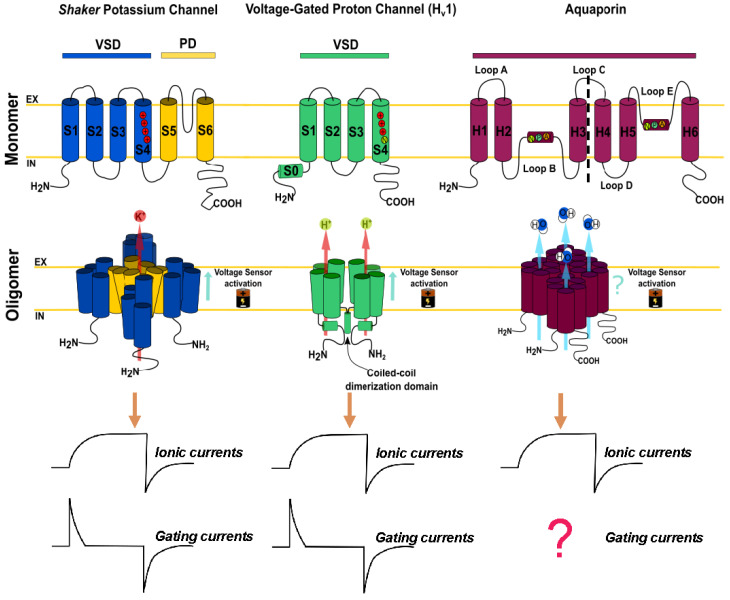
**Voltage-gated ion channels and their comparison with aquaporins.** The Shaker potassium channel and the Hv1 proton channel are members of the voltage-activated ion channel family. The Shaker potassium channel consists of monomers with six transmembrane segments (S1–S6), which possess two functional domains: the voltage sensor domain (VSD) surrounding the transmembrane segment 1 to 4 (S1–S4) and the pore domain (PD) ranging from transmembrane segment 5 to 6 (S5–S6). The VSD contains four positively charged residues (R) that confer voltage sensitivity. This protein is functionally expressed in the membrane, forming tetramers where the PD of each subunit constitutes a functional pore in the center of the oligomer. On the other hand, the Hv1 channel is a selective proton channel with four S1–S4 transmembrane segments, where the permeation pathway and the voltage sensor are in the same structural domain. The Hv1 voltage sensor consists of only three R residues. Although this protein naturally ensembles as a dimeric channel, each of its monomers is perfectly functional. The aquaporins are a large family of transmembrane proteins made up of six transmembrane segments (H1–H6) with two loops immersed into the membrane (Loop B and E). Each of these loops contains the NPA region, which is key to water conduction. In general, the selectivity filter for aquaporins consists of R and H residues. It has been observed in molecular dynamics simulations that these residues suffer conformational changes upon membrane depolarizations. This suggests that the movement of these residues can be experimentally recorded by electrophysiological techniques, just as it has been previously recorded in the *Shaker* channel and the Hv1 channel.

**Figure 3 ijms-23-12317-f003:**
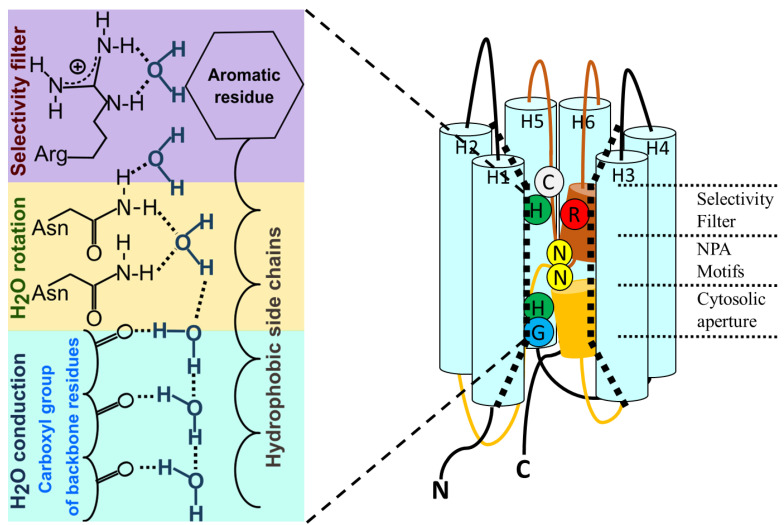
**General conduction mechanism in water-transporting aquaporins.** In water-selective AQPs, molecules permeate through the channel in a single file in a tremendously efficient way. This is achieved by the fact that one of the pore walls is made up of hydrophobic side chains, which decrease the friction between the pore and the permeating molecules, while on the other side of the conduction pathway, each molecule is coordinated through hydrogen bonds with the carboxyl groups of amino acids. When the water molecule reaches half of the way in the single file region, it faces the two N residues of the NPA motifs. Both of these N residues coordinate the molecule orientation by rotating the angle of the dipole moment of the water molecule, thus generating an energy barrier to the passage of protons. The conduction pathway of the AQPs has an hourglass-like shape where the narrowest part corresponds to the selectivity filter (SF). The SF consists of an R and an amino acid with an aromatic side chain, commonly H or F (Ar/R region). This filter blocks the passage of solutes larger than water. In some AQPs, there is a C residue next to the selectivity filter. This C is sensitive to HgCl_2_ blockage in some aquaporins, such as AQP1, or it is involved in the activation of ion conductance in AQP6 [25]. In the cytosolic aperture at the end of the single file region there are H-bond donor residues. The H95 located in this region is responsible for inhibition of water permeation by pH in human AQP4 [26]. In addition, G103 is a key residue for water and ion permeation through AtPIP2;1 [27]. For easy visualization, the intracellular loop B and the extracellular loop E are shown in color.

**Figure 4 ijms-23-12317-f004:**
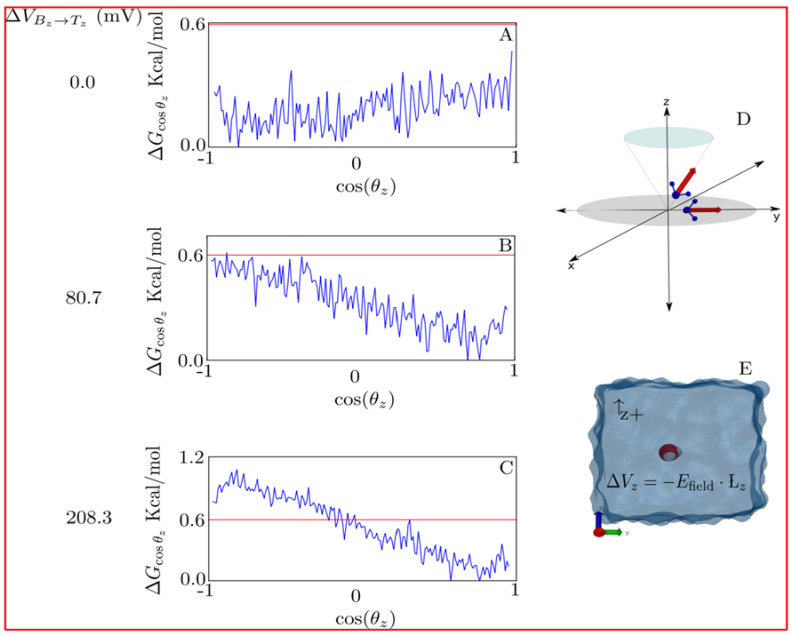
**Free energy of water dipolar orientation along +z direction (Δ*G_cosθz_*).** (**A**–**C**) Free-energy profiles (blue lines) under 0.0, 80.7 and 208.3 mV voltage differences (ΔV*_Bz→Tz_*), respectively. In red, thermal energy (kBT). (**D**) Scheme of the cosine of the angle between the dipole moment of the water molecule (red arrows) and the +z axis (cos(*θ_z_*)). (**E**) Simulated water box; solvent molecules (water) are represented in light blue. Red (oxygen) and white (hydrogen) spheres stand for the central water molecule for which Δ*G* estimations were computed. Calculations were carried out in NAMDv2.12 for single TIP3 water in a 24 × 24 × 24 water box under periodic boundary conditions, with a constant electric field (E) applied in the +z direction. For more details, see [171].

**Figure 5 ijms-23-12317-f005:**
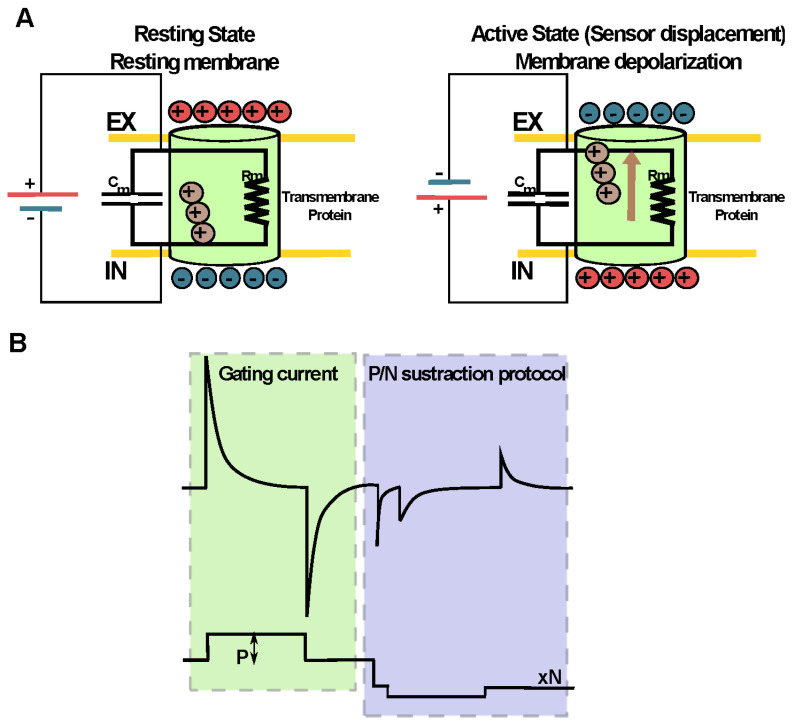
The conformational changes of a potential sensor can be studied directly through electrophysiological techniques. (**A**) The movement of charged particles produces a current. Thus, the movement of the side chains of the charged amino acids of the voltage sensor produces a characteristic transient current known as the gating current. These currents are a nonlinear capacitive component directly detectable by the voltage-clamp technique. (**B**) To separate the linear capacitive current originated by the passive membrane properties from the nonlinear capacitive currents that emerge from the motion of the sensor, a subtraction voltage protocol is used. This protocol, usually called P/N, consists of applying “N” voltage pulses of contrary polarity to the depolarizing pulse (P) used to test the displacement of the VSD. The magnitude of these N pulses results from dividing the test pulse by N, and the sum of the amplitudes of these “N” pulses must be equal to that of the test pulse P. Thus, the linear (Ohmic) component is withdrawn, and then the non-linear component that corresponds to the charge displacement of the VSD can be visualized.

**Figure 6 ijms-23-12317-f006:**
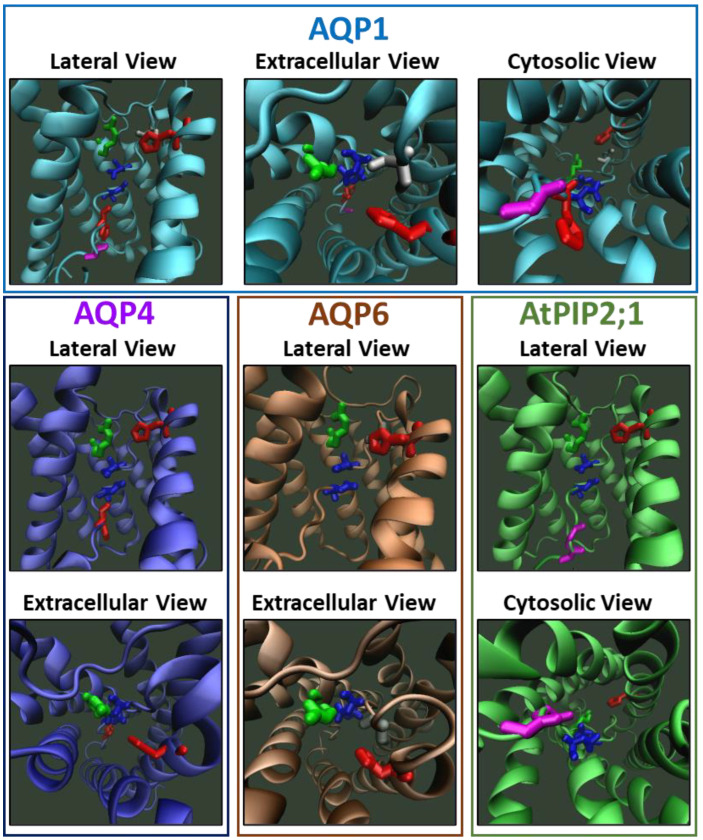
Conserved and key residues in the single-file water pathway of AQPs could be involved in the development of gating currents. Molecular representations of AQP1, AQP4 and AtPIP2;1 were developed on the bases of the PDB files 1J4N [58], 3GD8 [63] and 5I32 [84], respectively. The AQP6 model was built by homology modelling utilising the Swiss-Model (https://swissmodel.expasy.org/, 15 September 2022) public server [199,200] and the PDB file 1H6I [52] as templates. In all lateral views, the second transmembrane segment was hidden for better visualization of the water pathway. In all cases, both Asn residues of the NPA region are shown in blue, the His of the selectivity filter (SF) is shown in red and the Arg of the SF is shown in green. In addition, other important residues are indicated in each AQP. In AQP1, the extracellular Cys sensitive to HgCl_2_ inhibition is shown in light grey (just next to the SF), and His and Gly from the H-bond donor region at the cytoplasmic opening are shown in red and magenta, respectively. In AQP4, the His sensitive to intracellular pH acidification [26] is shown in red. In AQP6, the Cys involved in ion conductance [25] is shown in dark grey (just next to the SF). In AtPIP2;1, the Gly involved in ion conductance [27] is shown in magenta (in the H-bond donor region of the cytoplasmic mouth). The single file region is defined between the SF and the H-bond donor region located at the cytoplasmic opening (see Figure 3). According to molecular dynamic simulations with human AQP4, the His and the Arg of the SF are candidates to be involved in the development of gating currents in AQPs [170]. All these model representations were built only with the purpose of visualization and do not have further claims. This image was made with VMD v1.9.4a48 software support. VMD is developed with NIH support by the Theoretical and Computational Biophysics group at the Beckman Institute, University of Illinois at Urbana-Champaign.

**Table 1 ijms-23-12317-t001:** AQP crystal structures. Unrevealing the gating mechanism requires atomic resolution models of the end state in order to define the sequence of transitions that connect the open and closed states.

AQP	Species	Res. (Å)	PDB	Comments	References
**AQP0**	*Bos taurus*	1.9 2.4 2.5 2.24 7.01 25	2B6O 2B6P 3M9I 1YMG 2C32 3J41	✓ Association mode between AQP0 tetramers from juxtaposed membranes in the eye lens ✓ CaM regulates multimeric channels by facilitating cooperativity between adjacent subunits	[50,54,55,56]
**AQP1**	*Homo sapiens* *Bos taurus*	3.8 3.7 2.2 3.28 3.54	1FQY 1IH5 1J4N 4CSK 1H6I 6POJ	✓ Structural determinants for water permeation are extensively described	[14,52,57,58,59]
**AQP2**	*Homo sapiens*	2.75 3.7 3.05	4NEF 4OJ2 6QF5	✓ Trafficking capacity from intracellular vesicles to the plasma membrane	[60,61]
**AQP4**	*Rattus* *Homo sapiens*	3.2 2.8 1.8 10	2D57 2ZZ9 3GD8 3IYZ	✓ Array formation and cell adhesion ✓ Water conductance mechanism ✓ Role of the N terminus of AQP4 in the stabilization of orthogonal arrays	[62,63,64,65]
**AQP5**	*Homo sapiens*	2.2 2.6 3.5	3D9S 5C5X 5DYE	✓ Multiple phosphorylation sites	[66,67]
**AQP7**	*Homo sapiens*	2.2 1.9 3.99 3.70	6QZJ 6QZI 6N1G 6KXW	✓ Structure of AQP7 bound to glycerol	[68,69,70]
**AQP10**	*Homo sapiens*	2.3	6F7H		[71]
**AQPM**	*Methanothermobacter marburgensis* *Archaeoglobus fulgidus*	1.68 2.3 3.0	2F2B 2EVU 3NE2	✓ A subdivision between water-selective aquaporins, and water-plus-glycerol-conducting aquaglyceroporins	[72]
**GlpF**	*Escherichia coli*	2.2 2.8 2.1 2.7	1FX8 1LDA 1LDF 1LDI	✓ Primary permeant substrate glycerol	[73,74]
**AQPZ**	*Escherichia coli*	2.5 3.2 2.2 2.3 2.55 2.4	1RC2 2ABM 2O9D 2O9E 2O9F 3NK5	✓ Two distinct Arg-189 conformations associated with water permeation	[75,76,77,78]
**PfAQP**	*Plasmodium falciparum*	2.05	3C02	✓ The two NPA regions bear substitutions to Asn-Leu-Ala (NLA) and Asn-Pro-Ser (NPS), might participate in preserving the orientation of the selectivity filter asparagines in the center of the channel	[79]
**AQY1**	*Pichia pastoris*	1.15 1.4 0.88 1.3	2W2E 2W1P 3ZOJ 5BN2	✓ Proposed gating mechanism regulated by a combination of phosphorylation and mechanosensitivity. ✓ Ultra-high resolution allowed hydrogens to be modelled inside the water-conducting channel.	[80,81]
**PIP2;1**	*Spinacia oleracea*	2.1 (close) 3.9 (open) 2.3 2.95 2.05	1Z98 2B5F 3CLL 3CN6 3CN5	✓ Molecular dynamics simulations of the initial events governing gating ✓ S115E and S274E single SoPIP2;1 mutants and the corresponding double mutant ✓ Crystal structure of the AQP at low pH, reveals for the first time the structural basis for how this pH-sensitive histidine helps to keep the aquaporin in a closed state	[35,38,82]
**PIP2;4**	*Arabidopsis thaliana*	3.7	6QIM		[83]
**TIP2;1**	*Arabidopsis thaliana*	1.18	5I32	✓ Ammonia-permeable aquaporin ✓ Extended selectivity filter with the conserved arginine of the filter adopting a unique unpredicted position	[84]

**Table 2 ijms-23-12317-t002:** Mammalian AQP mutants and their impact on disease.

AQP	Mutations	Comments	References
**AQP0**	Ser235Ala	Produces defects to the plasma membrane translocation that induce congenic cataracts	[122]
	Arg33Cys	Reduces the cell-to-cell adhesion that is critical for lens transparency and homeostasis, inducing congenital lens cataracts	[129]
**AQP1**	Thr157Ala Thr239Ala	Abolish both the water permeability and the cationic conductance increase mediated by PKC phosphorylation in *Xenopus* oocytes	[130]
		Impair hypotonicity-induced translocation of AQP1 to the plasma membrane of HEK cells	[131]
**AQP2**	Cys181Trp	Impairs the plasma membrane translocation and the mutant accumulates in the endoplasmic reticulum (ER). This mutation was observed in a patient with congenital nephrogenic diabetes insipidus (NDI)	[132]
	Ser256Asp	Phosphorylation of serine 256 allows AQP2 trafficking to the plasma membrane. Mutations in this site abrogates the protein translocation. Thus, these mutations produce NDI, due the inability of the kidneys to concentrate urine	[133]
	Ser256Ala		
	Gly100Val	Both mutations were found in a Chinese family with congenital NDI, and both mutations retain AQP2 in the ER	[134]
	Gln57Pro		
	Ala147Thr	These three mutations were found in patients with NDI, impair translocation to the plasma membrane, and the mutant is retained at the ER	[135]
	Thr126Met		
	Asp68Ser		
	Arg254Leu	This mutation lacks basopressin-mediated phosphorylation at S256, causing impaired transport to the plasma membrane. This mutation also causes NDI disease	[136]
**AQP3**	Tyr19Ala	Partial disruption of AQP3 basolateral localization in MDCKII cells. A portion of AQP3 remains in the cytoplasm	[137]
	Leu21Ala-Leu22Ala		
	Tyr19Ala-Arg20Ala-Leu21Ala-Leu22Ala	The mutation of this N-terminal motif produces complete retention of AQP3 in the cytoplasm of transfected MDCKII cells	
**AQP4**	Ser180Ala	Ser180 is a PKC phosphorilation site. This mutation impairs the effect and produces a reduced water-permeable protein.	[138]
	Ser111Ala	Abolishes PKG-mediated phosphorylation of AQP4 in astrocytes cultured in vitro	[139]
	Ser276Asp	This point mutation increases the rate of protein degradation. The effect was observed in AQP4-transfected MDSCK cells lysates	[140]
**AQP5**	Ser156Glu	This phosphomimetic mutant increases the constitutive expression of AQP5 on the membrane but does not cause significant structural changes to the protein	[67]
**AQP6**	N-terminus	The short N-terminus of AQP6 is involved in its intracellular localization	[94]
**AQP7**	Ser10Ala/T11Ala	These mutations impair PKC phosphorylation of AQP7 and its interaction with perilipin 1 affecting the correct localization of AQP7 in the plasma membrane of adipocytes	[141]
	Gly264Val	Loss of glycerol transport function that impairs the reabsorption of glycerol by the kidneys inducing hyperglyceroluria in human children	[142]
**AQP9**	Ser11Ala	This unphosphorylated mutant cannot be localized in the plasma membrane	[143]
	Ser11Asp	This phosphomimetic mutant enhances neutrophil polarization and chemotaxis	
**AQP10**	His80Ala	Impairs glycerol permeability evidencing the role of His80 as a pH sensor	[71]
	Gly73Val	Impair glycerol permeability suggesting that the G73G74 motif mediates a gating mechanism between loop B and TM2 in the cytoplasmic aperture of the channel	
	Gly73Phe		
**AQP11**	Cys227Ser	Produces renal failure and death in sudden juvenile death syndrome (SJDS) in mice	[144]
		Alters the correct folding of AQP11, inducing endoplasmatic reticulum stress and apoptosis in cells of the proximal tubule	[145]

## Data Availability

Not applicable.

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
