# Peer review of "Aquaporin Gating: A New Twist to Unravel Permeation through Water Channels"

_ijms, 2022, doi:10.3390/ijms232012317_

Round 1

Reviewer 1 Report

In the present review, Ozu et al. proposed a very interesting and necessary update about Aquaporin gating. This is a central topic in the field of AQPs not largely studied, due to experimental limitations, and still unresolved.

In an attempt to open new challenges for future research, the authors have chosen a very clear and conducive way to play throughout the manuscript with the comparison of similarities and differences, in structure and mechanisms of gating, between ion channels and AQPs. They point out that although experimental evidence allows asserting that AQPs are gated by several mechanisms and molecular dynamics simulations predict the conformational changes related to those mechanisms, direct experimental evidence to confirm these events occurring in the core of the membrane plane is lacking. Finally, the authors propose that the displacement of key amino acid residues, located in the single file permeation pathway, is related to gating mechanisms and can be measured by means of the same methodology used on the proton channel. I believe that this proposal is original and amazing and opens new challenges for research in AQPs.

Minor points

1. There are some mistakes in section numbers. The correct should be:

 1. Introduction

2. Aquaporins: structure and function

2. pH gating in plant and animal aquaporins

3. pH gating in plant and animal aquaporins

3.1. The well-defined role of cytoplasmic loop D in plant PIPs

3.2. The role of histidines in animal aquaporins

3.3. The cytoplasmic loop B: more critical motifs

3.4. The ar/R selectivity filter and more charges available

4. Controversial issues in aquaporin research field: electrophysiological studies in ion conducting aquaporins (icAQPs)

5. How AQP is sensing the membrane electric field?  

6. Perspectives

2. Page 5, line 224. It should be better to replace the title of this section with: “Gating in plant and animal aquaporins” since there is a lot of information not only about the role of the pH in AQPs gating but also about evidence involving the putative role of membrane potential in this process.

Author Response

R#1: There are some mistakes in section numbers.

Answer: The numbers of the sections were corrected, and a new section was added following the suggestion from Reviewer #2.

R#1: Page 5, line 224. It should be better to replace the title of this section with: “Gating in plant and animal aquaporins” since there is a lot of information not only about the role of the pH in AQPs gating but also about evidence involving the putative role of membrane potential in this process.

Answer: Done

Reviewer 2 Report

Marcelo Ozu and colleagues submitted a Manuscript entitled “Aquaporin gating: a new twist to unravel the permeation through water channels” for publication as review in IJMS.  The current review manuscript discussed exclusively about the Aquaporin structure and gating function, molecular mechanism of gating by pH. Also, authors focused on the different AQP motifs and how AQPs sensing the membrane electric field.  The current manuscript is extensively informative; however, several issues raise concerns, needs to be addressed.  

Comments:

11)  It would be additional if authors could discuss how the aquaporins expression can be regulated intracellularly (trafficking) and what signaling mechanism involved and factors involved in the regulation of AQPs?

22)    Also, summary of table representing the evidence of mutations in different isoforms AQPs and diseases associated with those mutations and gain or loss of function of AQPS. These mutations lead to dysregulated expressions of AQPs and their functional significance.  

Author Response

R#2: It would be additional if authors could discuss how the aquaporins expression can be regulated intracellularly (trafficking) and what signaling mechanism involved and factors involved in the regulation of AQPs?

Answer: A new section was added according to the reviewer’s suggestion

R2: Also, summary of table representing the evidence of mutations in different isoforms AQPs and diseases associated with those mutations and gain or loss of function of AQPS. These mutations lead to dysregulated expressions of AQPs and their functional significance.

Answer: As stated before, we included a new section about trafficking and mentioned some mutations related to phosphorylation in AQPs. We also added a new table summarizing this issue in mammal AQPs.

Reviewer 3 Report

Aquaporin gating: a new twist to unravel the permeation through water channels.

This paper intents to focus on the gating of Aquaporin a phenomenon that has been described for Aquaporins of different origins. The major problem issued by the authors is to explore if gating currents lead to conformational changes (as stated in the line 33 in the abstract).

After an introduction the paper gives an overview of Aquaporin structure-function relationships followed by the molecular mechanism of pH gating in plant and animal aquaporins. The last third is dedicated to what the authors call "Controversial issues in aquaporin research: electrophysiological studies in ion conducting aquaporins /icAQPs)".

Aquaporins have regularly been the target of reviews, focusing on everything from structure-function relationship to physiological roles.

I believe the focus of the present review is relevant but I think the authors should consider expanding the "controversial" part and reducing the introductory part and the structure-function part as these subjects have been reviewed extensively many times before and nothing new is presented in these parts.

In addition to this I have a number of specific comments to the use of the English language. I believe the paper can be improved considerable by a thorough review of each sentence. The paper needs a thorough rewriting to clarify the presentation.

Many specific comments to the presentation are given below.

Line 21 and 22: A very long sentence that should be separated in two.

Line 29 and 30: Something is missing in the sentence, “both sustaining that the rearrangement of conserved residues contributes to occlude of the cavity of the channel restriction water permeation”. Is water permeation pH regulated?

Line 33: What is meant with a gating current? Do you mean an electrical current or what?

Line 41: I know people often use the term transport in relation to Aquaporins, but Aquaporins do not transport anything. They mediate a flux of either water or other small molecules in response to a chemical gradient. So I thing one should avoid using transport to emphasize this fact.

Line 53: Remove etc at the end of the sentence as it makes no sense.

Line 64: Rewrite the last part of the sentence, “which consisted in the overexpression of ….”. I would write “based on overexpression of ……”.

Line 70: Instead of “leads the water transport” I would write “mediates the water flux in RBD”

Line 72: Remove etc.

Line 71 to 74: I would change the sentence to “ many research groups initiated characterization of Aquaporins by cloning their genes from multiple species representing all kingdoms”. I would furthermore divide the very long sentence in two.

Line 74 to 77:  This sentence is difficult to comprehend. I would remove “other techniques” as this formulation is in contrast to the beginning of the sentence saying –mainly-. What do you mean by “ a huge amount of evidence has been collected” ? Evidence for what? And the sentence “ which to date has made it possible to partially answer the questions referred above”. What are the questions referred above?

Line 80 to 83: I think the authors should refer not just to Fig. 1 but also Fig. 2 and 3 as you for instance cannot see from Fig. 1 that the aromatic-arginine (a/R) motic faces the extra cellular vestibular region. Consider including a figure showing the molecular structure of AQP structure in more detail. This would help visualize the message in this part of the text.

Line 95: Something is missing in the last part of the sentence “in AQPs seems to be mandatory”. I furthermore don’t thing this statement is correct. It has been shown that mutants resulting in monomeric AQPs show water flux. (Victoria Schmidt  and James N. Sturgis ACS Omega  2017, 2, 6, 3017-3027 and
Borgnia, M. J.Kozono, D.Calamita, G.Maloney, P. C.Agre, P.  J. Mol. Biol. 19992911169 1179 DOI: 10.1006/jmbi.1999.3032 ).

Lines 100 - 101: I would change the first part of the sentence to “Thus, while ion channels can be characterized by electrophysiological methods” and rewrite the second part to make it more informative. It is not apparent what you mean with “the water transport capacity of AQPs relays on complementary biophysical approached”.

Line 102- 103: What is meant with “This is a difference of study capacities between ion and water channels”?

Line 104: Include “the” in front of single channel recordings.

Line 111: Change pH and Ca2+ to pH or Ca2+.

Line 117 to 119: The sentence “Certain plant AQPs-PIP subfamily- when expressed as heterotetramers-impact by increasing membrane water permeability if compared with homotetrameric forms” is unclear and should be altered. Suggestion: “Certain plant AQP-PIP subfamilies increase water permeability when present as heterotetramers relative to the homotetramers”.

Line 129: The sentence “ This plus the relative small size of AQPs make possible to perform molecular dynamics simulations with high predictive power” is unclear and I would suggest to reformulate something like this “ This plus the relative small size of AQPs sustains molecular dynamics simulations with high predictive power”.

Line 134: Remove s from AQPs.

Line 147: “The discussion will be focused in aquaporins” should be changed to “The discussion will be focused on aquaporins”.

Line 199: “water permeation through a single AQP cannot be yet unequivocally measure by” should be changed to “ water permeation through a single AQP cannot yet be unequivocally measured by”

Line 215 – 216: In the last part of the sentence “impact in” should be changed to allows.

Line 219: In the Bracket (o interweaved) should be changed to (or interweaved).

Line 220: “if we seek for what it has not been yet measured” What is meant by the sentence?

Line 232: Remove “s” from favors.

Line 236: remove “out” from the sentence.

Line 280: “By other side” should be changed to “on the other hand”

Line 285: xenopus oocytes should be changed to Xenopus oocytes.

Line 298: “not constitute α-helix should read “not constitute an α-helix”.

The 2.4 part describes a number of important residues. Comprehending this information would benefit a lot from a Figure showing the location of the mentioned residues.

Line 398: “Evidence revisited up to here give rise to ….” Should be changed to Evidence revisited up to here gives rice to ….”

Line 425: You miss an “an” before other article.

Line 595 in the legend to Figure 3: Something is missing in the sentence starting with “When the water molecule……….”

Author Response

R3: I believe the focus of the present review is relevant but I think the authors should consider expanding the "controversial" part and reducing the introductory part and the structure-function part as these subjects have been reviewed extensively many times before and nothing new is presented in these parts.

Answer: We agree with the reviewer’s comment and this section was rewritten and expanded to better expose our manuscript proposal. 

R3: In addition to this I have a number of specific comments to the use of the English language. I believe the paper can be improved considerable by a thorough review of each sentence. The paper needs a thorough rewriting to clarify the presentation.

Answer: the language was reviewed, and some parts of the text rewritten for clarification.

R3: Line 21 and 22: A very long sentence that should be separated in two.

Answer: Done

R3: Line 29 and 30: Something is missing in the sentence, “both sustaining that the rearrangement of conserved residues contributes to occlude of the cavity of the channel restriction water permeation”. Is water permeation pH regulated?

Answer: Done

R3: Line 33: What is meant with a gating current? Do you mean an electrical current or what?

Answer: We completely agree with the reviewer’s concern, for this reason, we can confidentially share some preliminary data with the reviewer (see attachment). Yes, we mean the gating current from AQP charges amino acid residues movement! The gating current is an electrical signal produced by the result of the displacement of charged residues or dipoles (voltage sensor domain) on the protein-channel. It is related to the conformational changes that occur when the channel transits from closed to open state. Now, we are preparing another manuscript to determine the voltage-dependent gating mechanism in different aquaporins and its relationship with water. 

We were recording the gating current in aquaporins: when the charged amino acids move, they will elicit a gating current upon depolarization, gating current that can be recorded by patch-clamp in X. laevis oocytes. 

We identified a potential coupling between the voltage sensor of AQP and the water displacement: the gating current pattern should change according to water permeability; in other words, the water flux is directly coupled with the voltage sensor movement in AQP. This allows conceptually to reformulate aquaporin as an “aqua-channel”.

The text of our manuscript was modified to clarify this issue.

R3: Line 41: I know people often use the term transport in relation to Aquaporins, but Aquaporins do not transport anything. They mediate a flux of either water or other small molecules in response to a chemical gradient. So I thing one should avoid using transport to emphasize this fact.

Answer: Done

R3: Line 53: Remove etc at the end of the sentence as it makes no sense.

Answer: Done

R3: Line 64: Rewrite the last part of the sentence, “which consisted in the overexpression of ….”. I would write “based on overexpression of ……”.

Answer: Done

R3: Line 70: Instead of “leads the water transport” I would write “mediates the water flux in RBD”

Answer: Done

R3: Line 72: Remove etc.

Answer: Done

R3: Line 71 to 74: I would change the sentence to “many research groups initiated characterization of Aquaporins by cloning their genes from multiple species representing all kingdoms”. I would furthermore divide the very long sentence in two.

Answer: We agree with the reviewer’s suggestion, and now the idea was separated into two sentences.

R3: Line 74 to 77: This sentence is difficult to comprehend. I would remove “other techniques” as this formulation is in contrast to the beginning of the sentence saying –mainly-. What do you mean by “a huge amount of evidence has been collected”? Evidence for what? And the sentence “which to date has made it possible to partially answer the questions referred above”. What are the questions referred above?

Answer: We agree with the reviewer and the text was modified for clarification and “other techniques” was removed.

R3: Line 80 to 83: I think the authors should refer not just to Fig. 1 but also Fig. 2 and 3 as you for instance cannot see from Fig. 1 that the aromatic-arginine (a/R) motic faces the extra cellular vestibular region. Consider including a figure showing the molecular structure of AQP structure in more detail. This would help visualize the message in this part of the text.

Answer: We referred to Figures 2 and 3, following the reviewer’s advice. This modification made the message clearer. 

R3: Line 95: Something is missing in the last part of the sentence “in AQPs seems to be mandatory”. I furthermore don’t thing this statement is correct. It has been shown that mutants resulting in monomeric AQPs show water flux. (Victoria Schmidt and James N. Sturgis ACS Omega 2017, 2, 6, 3017-3027 and Borgnia, M. J.; Kozono, D.; Calamita, G.; Maloney, P. C.; Agre, P. J. Mol. Biol. 1999, 291, 1169– 1179 DOI: 10.1006/jmbi.1999.3032).

Answer: AQP monomers have not been reported in nature. The two articles suggested by the reviewer regarding this matter are addressed to artificial systems (liposomes). 

The article from Borgnia et al. (1999) demonstrates that AQPs exist as tetramers in the membrane. In this publication, the authors try to develop a method to improve the expression and purification of AQPs. They used the AQPZ from E. coli and observed that the tetramer is very stable, even resistant to detergent treatments. They separated the tetramer in monomers (in vitro) by detergent incubation. The unitary permeability reported is calculated from stopped-flow experiments, which were performed in different protein/lipids proportions. The other article, Schmidt and Sturgis (2017) design specific mutants of AQPZ in order to disrupt the tetramers. These mutants were expressed in E. coli, and their purification was used to perform the experiments (in vitro) by stopped-flow using reconstituted liposomes.

However, we truly appreciate and consider the reviewer´s comment.  The text was modified to better explain the native oligomerization states of AQPs and these two references were included.

R3: Lines 100 - 101: I would change the first part of the sentence to “Thus, while ion channels can be characterized by electrophysiological methods” and rewrite the second part to make it more informative. It is not apparent what you mean with “the water transport capacity of AQPs relays on complementary biophysical approached”.

 Answer: The text was modified for clarification.

R3: Line 102- 103: What is meant with “This is a difference of study capacities between ion and water channels”?

Answer: This aspect is related to the previous one. The text was modified in order to clarify this matter.

R3: Line 104: Include “the” in front of single channel recordings.

Answer: Done

R3: Line 111: Change pH and Ca to pH or Ca .

Answer: Done

R3: Line 117 to 119: The sentence “Certain plant AQPs-PIP subfamily- when expressed as heterotetramers-impact by increasing membrane water permeability if compared with homotetrameric forms” is unclear and should be altered. Suggestion: “Certain plant AQP-PIP subfamilies increase water permeability when present as heterotetramers relative to the homotetramers”.

Answer: Done

R3: Line 129: The sentence “This plus the relative small size of AQPs make possible to perform molecular dynamics simulations with high predictive power” is unclear and I would suggest to reformulate something like this “This plus the relative small size of AQPs sustains molecular dynamics simulations with high predictive power”.

Answer: Done

R3: Line 134: Remove s from AQPs.

Answer: Done

R3: Line 147: “The discussion will be focused in aquaporins” should be changed to “The discussion will be focused on aquaporins”.

Answer: Done

R3: Line 199: “water permeation through a single AQP cannot be yet unequivocally measure by” should be changed to “water permeation through a single AQP cannot yet be unequivocally measured by”

Answer: Done

R3: Line 215 – 216: In the last part of the sentence “impact in” should be changed to allow.

Answer: Done

R3: Line 219: In the Bracket (o interweaved) should be changed to (or interweaved).

Answer: Done

R3: Line 220: “if we seek for what it has not been yet measured” What is meant by the sentence?

Answer: This sentence was reformulated to clarify the idea.

R3: Line 232: Remove “s” from favors.

Answer: Done

R3: Line 236: remove “out” from the sentence.

Answer: Done

R3: Line 280: “By other side” should be changed to “on the other hand”

Answer: Done

R3: Line 285: xenopus oocytes should be changed to Xenopus oocytes.

Answer: Done

R3: Line 298: “not constitute α-helix should read “not constitute an α-helix”. The 2.4 part describes a number of important residues. Comprehending this information would benefit a lot from a Figure showing the location of the mentioned residues.

Answer: The text was modified according to the reviewer’s comments. In addition, Figure 3 was modified to improve the visualization of key residues. The new figure (Figure 6) was included to highlight the relevant position of crucial amino acid residues. Notice the single file region of the water pathway of animal and plant AQPs.

R3: Line 398: “Evidence revisited up to here give rise to ….” Should be changed to Evidence revisited up to here gives rice to ….”

Answer: Done

R3: Line 425: You miss an “an” before other article.

Answer: Done

R3: Line 595 in the legend to Figure 3: Something is missing in the sentence starting with “When the water molecule……….”

Answer: Done
